# High cellulose dietary intake relieves asthma inflammation through the intestinal microbiome in a mouse model

**Song Wen[1], Guifang Yuan[1], Cunya Li[2], Yang Xiong[3,4], Xuemei Zhong[5], Xiaoyu Li[6]\***

1 Department of the First Clinical Medicine, Chongqing Medical University, Chongqing, China, 2 Department of the Traditional Medicine, Chongqing Medical University, Chongqing, China, 3 Andrology Laboratory, West China Hospital, Sichuan University, Chengdu, China, 4 Department of Urology, West China Hospital, Sichuan University, Chengdu, China, 5 Department of Respiratory Endocrinology, School of Clinical Medicine, Chongqing Medical and Pharmaceutical College, Chongqing, China, 6 Laboratory of Innovation, Basic Medical Experimental Teaching Centre, Chongqing Medical University, Chongqing, China

\* 100398@cqmu.edu.cn

**Data Availability Statement:** All relevant data are within the manuscript and its Supporting Information files.

## Abstract

Numerous epidemiological studies have shown that a high dietary fiber intake is associated inversely with the incidence of asthma in the population. There have been many studies on the role of soluble dietary fiber, but the mechanism of action for insoluble dietary fiber, such as cellulose-the most widely existing dietary fiber, in asthma is still unclear. The current study investigated the outcomes of a high-cellulose diet in a mouse model of asthma and detected pathological manifestations within the lungs, changes in the intestinal microbiome, and changes in intestinal short-chain fatty acids (SCFAs) in mice. A high-cellulose diet can reduce lung inflammation and asthma symptoms in asthmatic mice. Furthermore, it dramatically changes the composition of the intestinal microbiome. At the family level, a new dominant fungus family *Peptostreptococcaceae* is produced, and at the genus level, the unique genus *Romboutsla*, *[Ruminococcus]_torques_group* was generated. These genera and families of bacteria are closely correlated with lipid metabolism in vivo. Many studies have proposed that the mechanism of dietary fiber regulating asthma may involve the intestinal microbiome producing SCFAs, but the current research shows that a high-cellulose diet cannot increase the content of SCFAs in the intestine. These data suggest that a high-cellulose diet decreases asthma symptoms by altering the composition of the intestinal microbiome, however, this mechanism is thought to be independent of SCFAs and may involve the regulation of lipid metabolism.

## Introduction

Allergic asthma is a disease caused by allergens entering the respiratory tract and stimulating B lymphocytes to produce specific antibodies, mainly IgE, and stimulate mast cells, basophils, and other immune cells to release inflammatory mediators [1]. The main pathophysiological changes include chronic airway inflammation, airway hyper-responsiveness to various

**Funding:** This study were sponsored by Scientiific Research and Innovation Expriment of Chongqing Medical University, Grant Number: 201903. The grantees were Song Wen, Guifang Yuan and Cunya Li, I am the instructor of this project. This program was established by Chongqing Medical University to encourage undergraduate students to conduct clinical or basic scientific research and to exercise scientific thinking and research skills. So this project has no official website. The website of Chongqing Medical University was www.cqmu.edu.cn.

**Competing interests:** The authors have declared that no competing interests exist.

stimulus factors, namely Airway Hyper Reactivity (AHR) [2], infiltration of eosinophils and lymphocytes in the airway, degranulation of mast cells and goblet cell hyperplasia, the expression of Interleukin-4 (IL-4), IL-5, IL-13, Th2-dominated TNF, and the up-regulation of other cytokines [3].

Recently, many epidemiological studies have shown that the incidence of asthma is related to diet. A diet that is high in fat and low in fiber increases the risk of developing asthma [4]. Additionally, giving clinical asthma patients a diet high in fruits and vegetables can relieve their respiratory symptoms [5]. However, the specific mechanism of this beneficial clinical outcome is currently not understood.

Cellulose is an insoluble dietary fiber formed by the dehydration of many glucose molecules connected by β-1,4 glycosidic bonds. It is widely present in plant cell walls and is the most common dietary fiber in grains, vegetables, and fruits. Bacteria that can decompose cellulose are found in both the human [6] and mouse intestines [7]. Changes in cellulose intake can alter the composition of the intestinal microbiome of rats and the production of SCFAs [8], regulating the occurrence of intestinal inflammatory diseases by regulating lipid metabolism [9]. A low-cellulose diet has been reported to aggravate pathological symptoms in asthmatic mice [10]. Dietary cellulose can also reduce ozone-induced airway hyperreactivity [11].

Lipid metabolism can influence the polarization of immune cells and the development of inflammatory responses [12]. Abnormal lipid metabolism has been associated with the development of asthma [13, 14]. A high-fiber diet can reduce body weight and regulate dyslipidemia [15, 16] and is involved in the regulation of inflammation [17]. The hypothesis of microbial diet interaction [18] and related research [19] suggests that high-fiber diets can affect the occurrence and development of allergic diseases and other inflammatory diseases by changing the composition and metabolites of the intestinal microbiome. Most fiber-containing foods include approximately one-third of soluble fiber and two-thirds of insoluble fiber [20]. It is currently believed that most soluble dietary fiber is fermented and broken down by intestinal bacteria to produce short-chain fatty acids (SCFAs), which can participate in the initiation and development of asthma by reducing inflammation within the respiratory tract in asthmatic mice [21] and inhibit the migration of eosinophils [22]. However, the biological outcomes of dietary insoluble fibers such as cellulose are still unclear.

Therefore, to further explore the role of cellulose in the occurrence and development of asthma, we established a mouse model of asthma and intervened by giving high cellulose (30%) diet. After that, we determined the lung inflammation of mice by ELISA and H&E staining. The changes of intestinal microbiome and short-chain fatty acids in the feces of mice were detected. The results showed that the lung inflammation symptoms of asthmatic mice treated with high cellulose diet were significantly improved, and different dominant intestinal microbiome concerning lipid metabolism was produced, but this change may not be achieved by increasing the amount of short-chain fatty acids.

## Materials and methods

### Animals

A total of 30 SPF C57BL/6J mice, 8w, were provided by the Experimental Animal Center of Chongqing Medical University (SCXK-(Chongqing) 2018–0003, Chongqing, China). All experimental protocols were performed according to the Guide for the Care and Use of Laboratory Animals, and they were approved by the Institutional Ethics Committee of Chongqing Medical University. After one week of adaptive feeding under standard conditions, the 30 mice were divided into the normal group (N1-N10), asthma group (AS1-AS10), and high-fiber diet group (HF1-HF10). The normal and asthma groups were both fed with clean-grade feed

and purified water, while the high-fiber group was fed with high-fiber feed (Jiangsu Xietong Pharmaceutical Bio-engineering Co., Ltd.) and purified water. All three groups were free to eat and drink. The feed of the normal group contained 5% cellulose and the high-fiber feed contained 30% cellulose. The increased percentage of cellulose was achieved by reducing the amount of corn starch. The formula of the feed can be found in the S1 File.

The asthma group and the high-fiber diet group were injected intraperitoneally with the mixed preparation of V-grade OVA (Sigma, USA) and $Al(OH)_3$ (Alladin, CHINA), and stimulated by aerosol inhalation of grade II OVA (Sigma, USA) to prepare the mouse asthma model. On the 0th, 7th, and 14th days, mice were injected intraperitoneally with 0.2ml of the sensitizing solution, and on the 21st day, the challenge solution was inhaled by aerosol for 30 minutes each time for one week. The sensitization solution is a normal saline suspension with V-grade OVA 1mg/ml and Al (OH)$_3$ 200mg/ml. The stimulating fluid is a physiological saline solution of grade II OVA 10mg/ml. The normal group is not processed. All groups were sampled within 24h after the end of nebulization on the 28th day.

## Determination of IgE, IL-4 levels, and H&E staining

Enzyme-linked immunosorbent assay (ELISA) kits were used to determine IgE and IL-4 levels (MultiSciences Biotech Co., Ltd. Hangzhou, China), and the kit instructions were; 'take out the serum samples from the refrigerator and gradually return to room temperature, and then measure the serum IgE and IL-4 levels. After fixing the lung tissue with 10% formalin solution for 24 hours, it was embedded in paraffin, sectioned (3μm), and stained with HE. The infiltration of inflammatory cells in the lungs of mice was observed under a light microscope.

## DNA extraction, PCR Amplification, and sequencing

The frozen tissue samples were thawed at room temperature and then homogenized via bead beating (FastPrep bead matrix E, MP Biomedicals, Santa Ana, CA, USA) with 500 μl of aseptic saline. Aseptic saline was added until a volume of 1000 μl was obtained, and bacterial DNA was extracted with a bacterial genomic DNA extraction kit as per the manufacturer's instructions. After DNA concentration was determined, 1 μl of DNA was diluted to 100 ng/μl with ultra-pure water and then stored at −20˚C.

Universal primers 338F/806R were synthesized to amplify the 16s rRNAV4–V5 region. The sequences of the forward primer and reversed primer were 5′–ACTCCTACGGGAGGCAGCAG– 3′ and 5′–GGACTACHVGGGTWTCTAAT–3′, respectively. In brief, PCR consisted of 4 μl of 5× FastPfu Buffer, 2 μl of dNTPs (2.5 mM), 0.8 μl of forwarding primer (5μM), 0.8 μl of reverse primer (5 μM), 0.4 μl of FastPfu polymerase, 10 ng of template DNA, and ultra-pure water added to obtain a volume of 20 μl. The reaction conditions were as follows: 5 min at 95˚C, 27 cycles of 30 s each at 95˚C and 53˚C and 45 s at 72˚C, and 10 min at 72˚C. The amplification products were analyzed through Illumina high-throughput sequencing (Illumina PE250). Finally, microbial diversity was analyzed based on the sequencing results.

## Short chain fatty acids extraction and analysis

Place an approximately 50 mg sample of feces into a 2 ml grinding tube, add a steel ball, 450 μl of methanol, and 50 μl of internal standard (1000 μg/ml of 2-ethyl-butyric acid, methanol configuration), and grind it in a freezing grinder 50HZ 3 twice per minute. Then ultrasonic the sample in an ice-water bath for 30 minutes, stand at -20˚C for 30 minutes, and centrifuge at 13000g for 15 minutes (4˚C). Transfer the supernatant to a 1.5 ml centrifuge tube. Add 50mg of anhydrous sodium sulfate, vortex, centrifuge at 13000g for 15min (4˚C), and take the supernatant solution on the machine for gas chromatographic analysis. The analytical instrument

used is Agilent Technologies Inc. (CA, USA) 8890B-5977B GC/MSD GC/MSD. HP FFAP capillary column (30 m × 0.25 mm × 0.25 μm, Agilent J&W Scientific, Folsom, CA, USA). The protocol used included a carrier gas that is high-purity helium (purity not less than 99.999%), the flow rate is 1.0 ml/min, the inlet temperature is 260˚C, an injection volume of 1μl, split injection, split ratio 10:1, and solvent extension 3min. Program temperature rise: the initial temperature of the column oven is 80˚C, the temperature is programmed to increase to 120˚C at 40˚C/min increments, and then increased to 200˚C at 10˚C/min steps, and finally runs at 230˚C for 6 minutes. Mass spectra were collected using an electron impact ion source (EI), ion source temperature 230˚C, quadrupole temperature 150˚C, transmission line temperature 230˚C, and electron energy 70eV. The scanning mode is the full scan mode (SCAN), and the quality scan range: m/z: 30–300. The obtained data were assessed using Masshunter quantitative software (Agilent, USA, version number: v10.0.707.0) to automatically identify and integrate each ion fragment with default parameters, and assist manual inspection. Linear regression standard curve lines were drawn with the mass spectrum peak area of the analyte as the ordinate and the concentration of the analyte as the abscissa. Sample concentrations were calculated by substituting the mass spectrum peak area of the sample analyte into the linear equation to calculate the concentration results.

## Statistical analysis

The Prism software (GraphPad) was used for statistical analysis of the data. The data of cytokine were statistically analyzed by one-way ANOVA, and for the microbiome data, the data between different groups were detected by Student's-T test method and Tukey method. (*, P <0.05; **, P <0 .01; ***, P <0 .001).

## Results

### High-fiber diet can reduce lung inflammation and the production of IL-4 and IgE

Pathological sections of the lung showed that the walls and smooth muscles of the small bronchioles in group N were thin, with no foreign bodies in the cavities and alveoli. There was no obvious inflammatory cell infiltration around the bronchial tube or the blood vessel walls. In the AS group, the walls of small and medium bronchioles and blood vessels were thickened, smooth muscles were proliferated, and showed hypertrophy and the lumen was more narrow. Many eosinophils and lymphocytes were infiltrated around the bronchi and blood vessels. The lung interstitium and alveolar cavity are filled with exudate, and the alveolar space is widened. Compared with the AS group, the HF group had a relatively complete airway epithelial structure, and the degree of inflammatory cell infiltration around the bronchus and blood vessels was significantly reduced. Exudates in the lung interstitium and alveolar cavity are also significantly reduced (see Fig 1).

The levels of IL-4 and IgE in the AS group were statistically significantly higher than those in the N group (P<0.05, P<0.05), but the levels of IL-4 and IgE in the HF group were significantly lower than those in the AS group (P<0.01, 0<0.05).

### Microbial diversity analysis

**Alpha-diversity.** The student's T-Test analysis was applied to bacterial DNA and genomic data. The Shannon index, Simpson index, Ace index, Chao index, and Coverage index of each sample are shown in Table 1. The results showed that the Shannon index of the AS and HF groups were significantly different (P<0.05), however no difference in the Simpson indexes

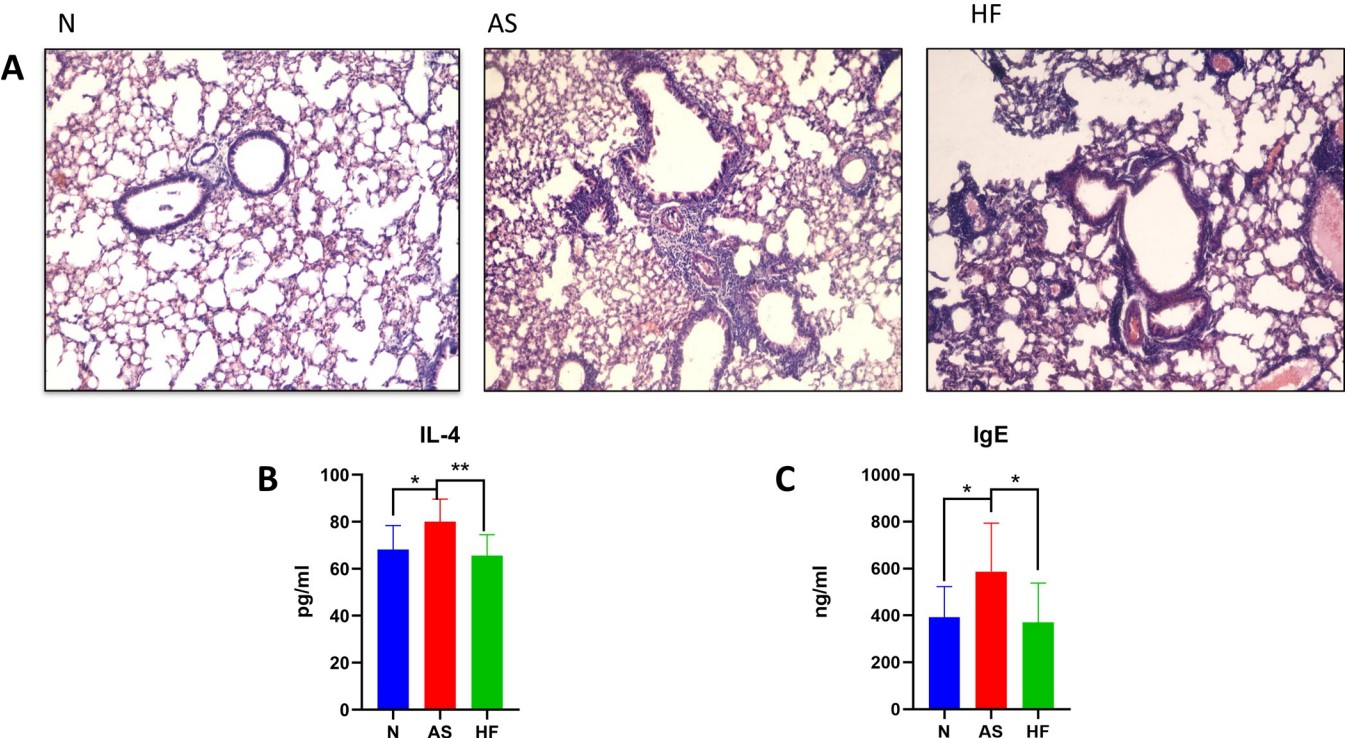

**Fig 1. High-fiber diet can reduce lung inflammation and related cytokine production.** A: The pulmonary inflammation in the AS group was significantly worse than that in the N group, but the situation was alleviated in the HF group. B: The IL-4 level in the AS group was significantly higher than that in the N group (P<0.05), while the IL-4 level in the HF group was significantly reduced (P<0.01). C: The IgE level in the AS group was higher than that in the N group (P<0.05), while the IL-4 level in the HF group was significantly reduced (P<0.05).

for each group was found. This indicates that there is a difference in community diversity between the AS and HF groups, while no difference between the N group and the other two groups was found. Both the Ace index and Chao indexes yielded significant differences between the N and HF groups (P <0.05, P <0.05), and between the AS and HF groups (P <0.001, P <0.001). These results confirm that there are significant differences in community richness between the HF group and those on lower nonsoluble fiber diets.

**Community-composition and beta-diversity.** Community barplot analysis showed that on the family level (see Fig 2), the dominant families in the AS group were *Prevotellaceae*, *Lachnospiraceae*, and *Muribaculaceae*; the dominant families in the HF group were *Lachnospiraceae*, *Peptostreptococcaceae*, and *Desulfovibrionaceae* and the dominant families in the N group were *Muribaculaceae*, *Prevotellaceae*, and *Bacteroidaceae*; The abundance of *Peptostreptococcaceae* in the HF group increased significantly, resulting in the emergence of a newly dominant family of bacteria.

**Table 1. The Alpha diversity analysis of microbiome in feces.**

|  | $N_{mean±sd}$ | $AS_{mean±sd}$ | $HF_{mean±sd}$ | $P_{AS-HF}$ | $P_{HF-N}$ | $P_{AS-N}$ |
|---|---|---|---|---|---|---|
| SHANNON | 3.7070±0.5590 | 3.7659±0.3632 | 3.2486±0.5736 | 0.0265 | 0.0860 | 0.7820 |
| SIMPSON | 0.0837±0.0649 | 0.0746±0.0279 | 0.1037±0.0695 | 0.2338 | 0.5123 | 0.6883 |
| ACE | 473.1872±74.7466 | 528.2429±77.3340 | 379.1904±78.4053 | 0.0001 | 0.0033 | 0.1033 |
| CHAO | 482.7851±84.5991 | 533.6014±80.4786 | 374.7840±66.6396 | 0.0001 | 0.0065 | 0.0846 |
| COVERAGE | 0.9979±0.0004 | 0.9978±0.0004 | 0.99855±0.0003 |  |  |  |

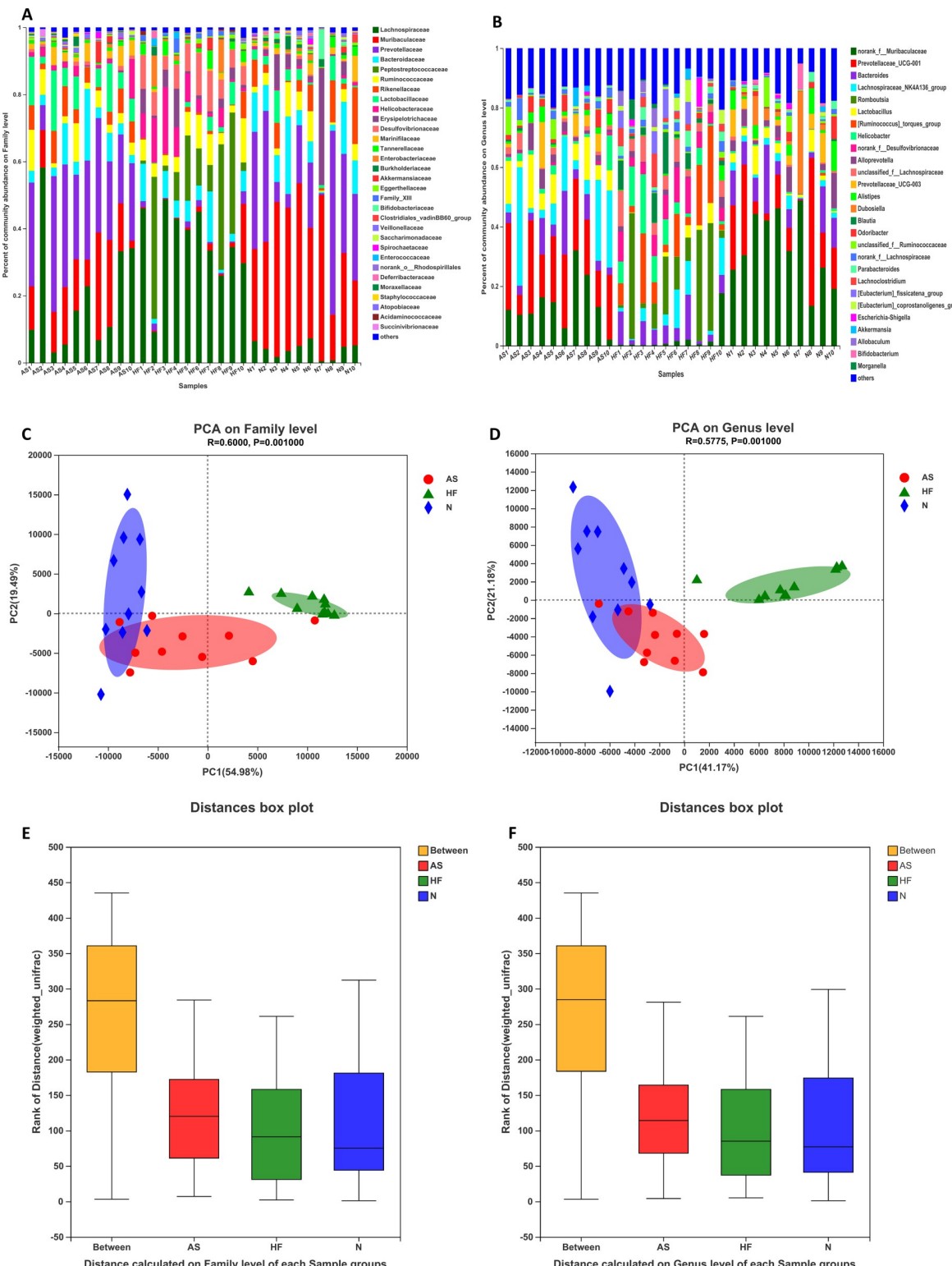

**Fig 2. Fecal microbial composition analysis and beta-diversity analysis.** A, B: analysis of the fecal microbial composition of the three groups on the family level and genus level; B, C: PCA analysis is used to explore the correlation of the composition of the microbiome on the family level and genus level of each group; C, D: ANOSIM analysis compares differences between groups and within groups (C:P <0.001, D: P<0.001).

At the Genus level (Fig 2B), compared with the N group, the proportion of *norank_f__Muribaculaceae* in the AS group decreased, while the proportion in the HF group decreased significantly. Again, compared to the N group, the proportion of *Lactobacillus* and *Lachnospiraceae_NK4A136_group* in the AS group slightly increased, while the proportion in the HF group decreased. Additionally, some uniquebacterial groups emerged in the HF group, such as *Romboutsla*, and the *[Ruminococcus]_torques_group*.

PCA analysis and ANOSIM methods were used to analyze bacterial populations at the family and genus levels, respectively. The PCA analysis showed that at the family level, the composition of the microbiome of the N group and the AS group overlapped, but the composition of the HF microbiome was significantly different (Fig 2C). The results of PC1 and PC2 were 54.98% and 19.49%. At the genus level, the N group and AS group also showed an inevitable overlap, while the HF group has a unique microbiome composition (Fig 2D). The data for PC1 and PC2 are 41.7% and 21.18%. ANOSIM results showed that at the family level (Fig 2E), the difference between groups was more significant than the differences within the groups (Weighted_Unifrac R = 0.7145, P<0.001). At the genus level (Fig 2F), the difference between the groups was also more significant than the difference within the groups (Weighted_Unifrac R = 0.7271, P<0.001)). These data show that three groups have statistically significant differences, regardless of whether the analysis is at the family level or the genus level.

## Analysis of significant differences between the groups at both the family and genus levels

As Fig 3 shows, at the family level, there are significant differences between the three groups in populations of *Lachnospiraceae*, *Muribaculaceae*, *Prevotellaceae*, *Peptostreptococcaceae*, *Rikenellaceae*, and *Lactobacillaceae* families. At the genus level, populations of *norank_f__Muribaculaceae*, *Prevotellaceae_UCG-001*, *Romboutsia*, *[Ruminococcus]_torques_group*, *Helicobacter*, and *norank_f__Desulfovibrionaceae* are significantly different in the three groups (P<0.001).

The Kruskal-Wallis test was then applied to conduct a multi-group analysis between *Romboutsia*, *[Ruminococcus]_torques_group*, and *Helicobacter*, which had noticeable compositional changes in the HF group at the genus level. *Romboutsia* is significantly different between the HF and AS groups, and the HF and N groups (P<0.01 and P<0.01 respectively), while *[Ruminococcus]_torques_group* shows significant differences between the HF and AS groups, and HF

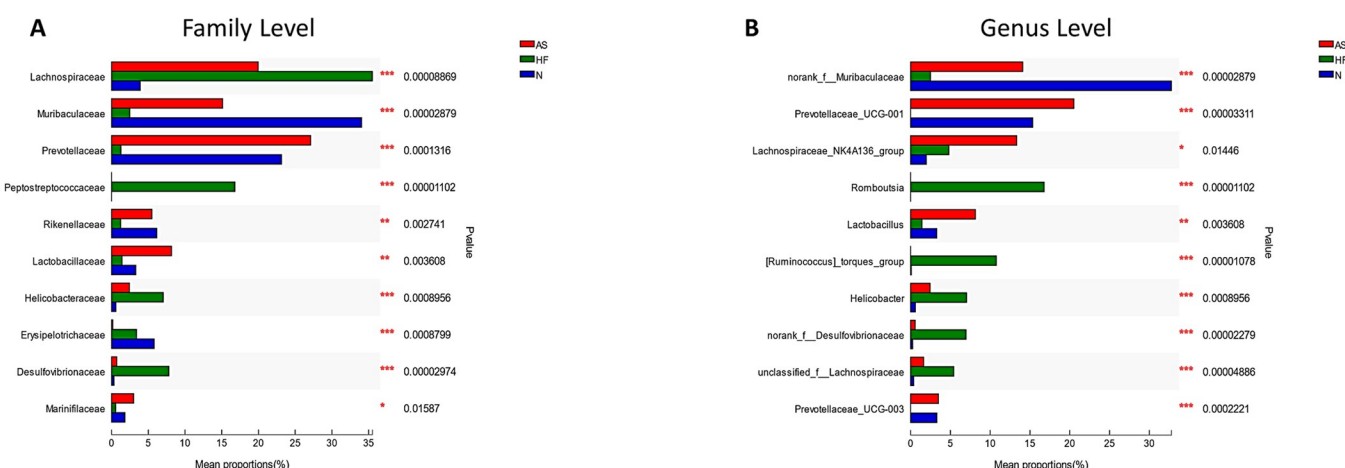

**Fig 3. Significantly different species analysis on the family level and genus level.** A: Analysis of significantly different species at the family level; B: Analysis of significantly different species at the genus level (*, P < 0.05; **, P <0 .01; ***, P <0 .001).

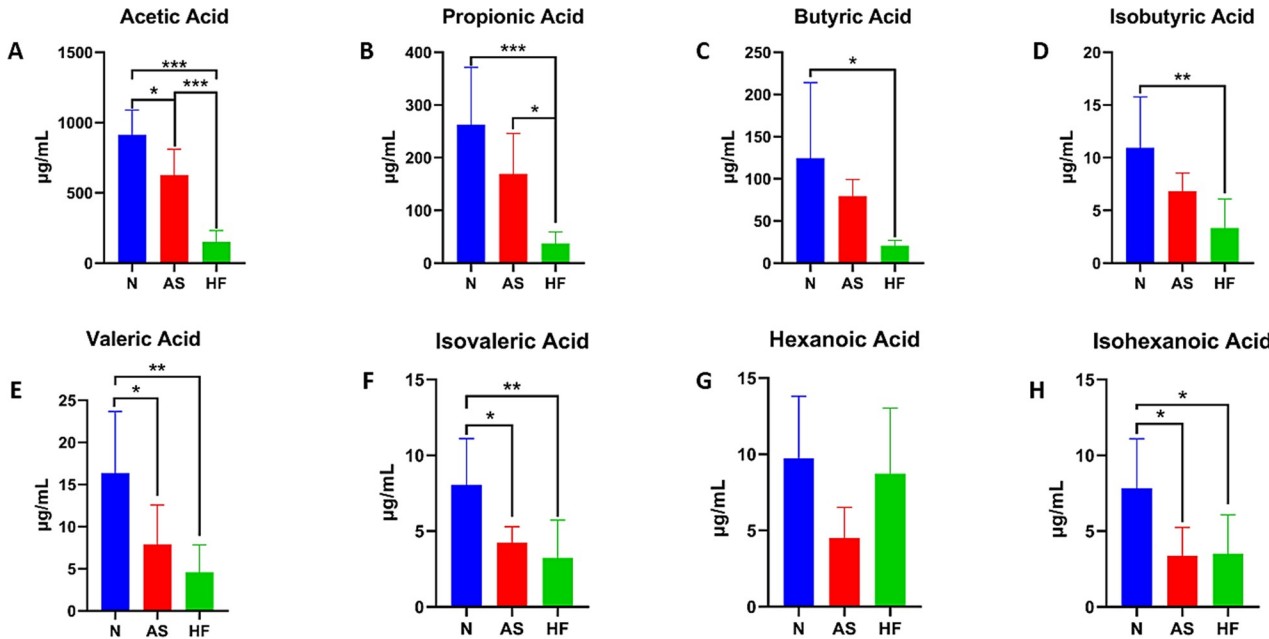

**Fig 4. Analysis of short-chain fatty acid content in feces.** A-H: Quantitative analysis of short-chain fatty acids in mouse feces.

and N groups (P<0.001 and P<0.001 respectively). Finally, the composition of *Helicobacter* bacteria is also significantly different between the HF group and AS groups and between the HF and N groups (P<0.01 and P<0.05 respectively).

**Short-chain fatty acid (SCFAs) analysis.** One proposed mechanism that the intestinal microbiome affects human health and disease is through the production of short-chain fatty acids in the intestine. We used GC-MS to detect the SCFAs content in mouse feces (Fig 4). Surprisingly, the contents of various SCFAs in feces from the HF group were lower than the other two groups. To analyze the correlation between the distribution of short-chain fatty acids and the intestinal microbiome, we performed a Variance inflation factor (VIF) analysis. The results showed that at the genus level, the distribution of the bacterial microbiome showed a strong positive correlation with acetic acid, isobutyric acid, and butyric acid, and a weak correlation with concentrations of isohexanoic acid, but no significant correlation with hexanoic acid.

## Discussion

Cellulose is the most common insoluble dietary fiber. In the current study, feeds containing different concentrations of cellulose were fed to asthmatic mice to study the effect of a high-fiber diet on lung inflammation in asthmatic mouse models. Also, the influence that varying amounts of dietary cellulose had on the microbiome of the mice intestinal tracts was delineated and analyzed.

The hypothesis of microbial diet interactions [18] proposes that cellulose may participate in the progression of allergic diseases by affecting the composition of the intestinal microbiome, which is closely related to pulmonary inflammation. Disrupting the composition of the intestinal microbiome with antibiotics can intensify the reaction of Th2 cells [21, 23], thereby intensifying the production of lung inflammation. Alterations in intestinal bacteria caused by the administration of antibiotics during the perinatal period have been reported to increase susceptibility to developing asthma in the future [24]. The data presented from the current study

shows that a high-cellulose diet reduces the total OUT (Operational Taxonomic Units) number of intestinal bacteria, while also producing more unique OTU sequences. In contrast, the HF group is lower than the AS group in community richness and community diversity. This is consistent with Kim Y's research results [9], however, contrasts with the conclusions from Peng X et al. study [8].

A principal mechanism that the intestinal microbiome affects remote organs is through the metabolism of SCFA's [25, 26]. Soluble dietary fiber can produce large quantities of SCFA's which then participate in the progression of inflammation and allergic diseases. However, our study has shown that insoluble cellulose cannot be fermented by intestinal bacteria to produce larger amounts of SCFAs. The concentrations of almost all SCFAs were significantly reduced (except for hexanoic acid). Similarly, Kim et al. (2020) showed that increased dietary cellulose intake did not result in changes to the intestinal acetic acid, propionic acid, and butyric acid contents.

*Romboutsla* and *[Ruminococcus]_torques_group* merits special attention. They are a unique microbiome induced by a high-fiber diet, present only in very small quantities in the N and AS groups. *Romboutsla* is classified as a member of the *Peptostreptococcaceae* family [27]. These bacteria primarily produce both acetic and lactic acids but also produce C16:0 saturated straight-chained fatty acids. It does not, however, use cellulose as a fermentation substrate. *[Ruminococcus]_torques_group* is classified as being in the *Lachnospiraceae* family [28], and its main fermentation products are acetic acid, propionic acid, and lactic acid. It is involved in the regulation of GPR41 stimulated by SCFAs, which induces adipocytes to produce leptin and influences the lipid profile. The *Lachnospiraceae* family is also significantly increased in the HF group and these bacteria activate the oxidation and new synthesis of fatty acids, inhibits lipid decomposition, thereby reducing circulating plasma lipid levels and body weight [29]. The data from Kim et al. (2020) showed that the intake of antibiotics can reduce the beneficial effects of cellulose on colitis. This study also demonstrated that cellulose regulates colon inflammation by regulating the metabolic spectrum of intestinal bacteria. The linoleic acid, nicotinate and nicotinamide, glycerophospholipid, glutathione, and sphingolipid metabolic pathways were all significantly modified by cellulose intake [9]. Tibbitt [30] performed single-cell RNA sequencing on TH2 cells in the airway and found that fatty acid oxidation and synthesis genes are highly enriched in TH2 cells. Simultaneously, blocking lipid metabolism pathways in vivo can inhibit T helper cell differentiation and the development of airway inflammation. These findings all support the proposal that the regulation of cellulose on the intestinal microbiome may not be through fermentation pathways that produce increased concentrations of SCFA's but through the regulation of lipid metabolism and other metabolic pathways. Importantly, this study has demonstrated that the mechanisms and outcomes that increased dietary insoluble fiber exert on the gut microbiome and overall inflammatory processes require further exploration and characterization.

Presently, most research investigating the effects of cellulose on inflammation focuses on intestinal inflammation. Nagy-Szakal et al. (2013) showed that supplementing cellulose in early life can alleviate the symptoms of colitis in mice [31]. This early study also demonstrated changes in intestinal bacteria but did not fully characterize or explain these results. Fischer et al. (2020) reported that cellulose may participate in the occurrence of colitis by regulating the function of intestinal immune and epithelial cells [7]. Whilst investigating cellulose and respiratory diseases, Zhang et al. (2016) reported that the intake of cellulose can improve OVA-induced pulmonary inflammation [10], focusing primarily on changes in cytokines in the alveolar lavage fluid. Tashiro et al. (2020) reported that cellulose can improve ozone-induced airway hyperresponsiveness, which is one of the main pathological features of asthma [11].

Generally, this research showed that a high dietary intake of insoluble cellulose was able to reduce lung inflammation in asthmatic mice. This outcome may not be induced by increased intestinal SCFAs, but rather by regulating the composition of the intestinal flora to regulate the body's lipid metabolism, thereby participating in the inflammatory response of remote organs.

## Supporting information

**S1 Table. Formula of high cellulose feed.**
(DOCX)

**S2 Table. Formula of regular feed.**
(DOCX)

**S1 File.**
(RAR)

## Author Contributions

**Conceptualization:** Song Wen, Cunya Li.

**Data curation:** Song Wen, Guifang Yuan, Yang Xiong.

**Formal analysis:** Guifang Yuan, Yang Xiong.

**Methodology:** Song Wen, Guifang Yuan.

**Project administration:** Xiaoyu Li.

**Resources:** Xuemei Zhong.

**Software:** Yang Xiong.

**Writing – original draft:** Song Wen, Xiaoyu Li.

**Writing – review & editing:** Xiaoyu Li.

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
