## [Decision Letter · Decision Letter 0]

19 May 2021

PONE-D-21-07750

High Cellulose dietary intake relieves asthma inflammation through the intestinal microbiome in a mouse model

PLOS ONE

Dear Dr. Li,

Thank you for submitting your manuscript to PLOS ONE. After careful consideration, we feel that it has merit but does not fully meet PLOS ONE’s publication criteria as it currently stands. Therefore, we invite you to submit a revised version of the manuscript that addresses the points raised during the review process.

More specifically, the reviewer 3 raised several points that need to be adressed in order to validate your final results. Please see the comments.

We look forward to receiving your revised manuscript.

Kind regards,

Catherine Mounier

Academic Editor

PLOS ONE

Journal Requirements:

2. Thank you for including your ethics statement:  "Experimental Animal Center of Chongqing Medical University [SCXK-(Chongqing) 2018-0003]".

Please amend your current ethics statement to confirm that your named ethics committee specifically approved this study.

For additional information about PLOS ONE submissions requirements for ethics oversight of animal work, please refer to http://journals.plos.org/plosone/s/submission-guidelines#loc-animal-research  

[We are grateful to The Innovation Lab,Chongqing Medical University.]

 [The authors received no specific funding for this work.]

5. We note you have included a table to which you do not refer in the text of your manuscript. Please ensure that you refer to Table 1 in your text; if accepted, production will need this reference to link the reader to the Table.

6. Please ensure that you refer to Figure 1, 2, 3 and in your text as, if accepted, production will need this reference to link the reader to the figure.

Reviewers' comments:

Reviewer's Responses to Questions

**Comments to the Author**

1. Is the manuscript technically sound, and do the data support the conclusions?

Reviewer #1: Yes

Reviewer #2: Yes

Reviewer #3: No

2. Has the statistical analysis been performed appropriately and rigorously? 

Reviewer #1: Yes

Reviewer #2: Yes

Reviewer #3: No

3. Have the authors made all data underlying the findings in their manuscript fully available?

Reviewer #1: Yes

Reviewer #2: Yes

Reviewer #3: Yes

4. Is the manuscript presented in an intelligible fashion and written in standard English?

Reviewer #1: Yes

Reviewer #2: Yes

Reviewer #3: No

5. Review Comments to the Author

Reviewer #1: Overall, the paper is methodologically sound, the statistical analyses that are presented seem adequate, and the interpretations reasonable. For some reason, messages such as "(see Fig Error! Reference source not found.)" are scattered throughout the text, please edit. Were the mice used male,female, both? If only male or female, why was one sex chosen, and how may that affect results? Why was the number of mice chosen used? Was a power analysis or other statistical analysis done to ensure proper numbers of mice?

Reviewer #2: The paper “High Cellulose dietary intake relieves asthma inflammation through the intestinal microbiome in a mouse model”, by Song Wen and colleagues, aimed to evaluate the role of a high-cellulose diet in a mouse model of asthma, detecting pathological manifestations in lungs, changes in the gut microbiome, and changes in intestinal SCFA in a murine model.

The paper is well written and the results clear to the reader.

Here you find my suggestions for this paper:

• Table 1: I think that the readability of this table could be increased as follow: mean and SD could be put together (mean�SD) and you should add the p-value. Moreover, you can add a legend for this table.

• Lines 165-167: you could also provide data for differences between N and HF.

• Line 48: the term “asth ation” is unclear. Please amend it.

• Many times (i.e. line 164, 181, 193, 206, etc.), in the paper there is reported the following “Error! Reference source not found”, in correspondence to references to Figures and Tables citations. Please amend it.

Reviewer #3: SEE THE FOLLOWING COMMENTS BELOW

Introduction:

a. Line 48, the reviewer is not familiar with asth ation, can the authors clarify what this word is and its meaning. Perhaps it is a typo.

b. The introduction provides information about some background, but it is jumbled and would benefit from a reorganization and rewrite. Further there does not seem to be an overt purpose or hypothesis established for the study.

Methods:

c. The mice need a more complete description and location of the vendor purchased from. Microbiome/microbiota can vary greatly between vendors and thus influence results of the study.

d. Given the perceived purpose is to examine the influence of fiber and asthma, the experimental design seems incomplete, there is no asthma-fiber group. Only normal, asthma and high fiber.

e. There does not appear to the tables for dietary composition. How were diets altered to account for the high cellulose content? Protein, fat? There was a lot of time spent on the importance of fat in the introduction. This would be critical for understanding interpretation of the results.

f. If all animals were challenged with OVA, then should the groups be renamed?

g. The authors do not appear to have a scoring outline in place to understand how the IHC data was analyzed to note differences in staining.

h. The authors have 16S rDNA in the V4-V5 regions, this should be rRNA. While one must extract the DNA it then must be prepared for rRNA analysis.

i. SCFA in animal models are analyzed using cecal contents so that fecal pellets can be used for microbiota.

j. The statistical analysis is very vague and does not employ appropriate techniques to analyze m data of this fashion.

Based on these concerns any results that followed would be impossible to interpret given the methodological fatal flaws.

6. PLOS authors have the option to publish the peer review history of their article (what does this mean?). If published, this will include your full peer review and any attached files.

Reviewer #1: No

Reviewer #2: No

Reviewer #3: No

---

## [Author Response · Author response to Decision Letter 0]

2 Jun 2021

Dear Editor and Reviewers:

Thank you for your letter dated May 19. We were pleased to know that our work was rated as potentially acceptable for publication in Journal, subject to adequate revision. We thank the reviewers for the time and effort that they have put into reviewing the previous version of our manuscript entitled “High Cellulose dietary intake relieves asthma inflammation through the intestinal microbiome in a mouse model”. Their suggestions have enabled us to improve our work. Based on the instructions provided in your letter, we uploaded the file of the revised manuscript. Accordingly, we have uploaded a copy of the original manuscript with all the changes highlighted by using the track changes mode in MS Word. Appended to this letter is our point-by-point response to the comments raised by the reviewers. The comments are reproduced and our responses are given directly afterward in a different color (red). We would like also to thank you for allowing us to resubmit a revised copy of the manuscript.

Responds to the reviewers' comments:

Reviewer #1:

a: Response to comments: Messages such as "(see Fig Error! Reference source not found.)" are scattered throughout the text, please edit.

Response: Thanks for your comment. We apologize for the incorrect writing errors in the manuscript, and we have corrected some of the spelling and formatting errors that have already been mentioned.

b: Response to comments: Were the mice used male, female, both? If only male or female, why was one sex chosen, and how may that affect results?

Response: Thanks for your comment. All the mice used in our experiments are male mice. The reason for this decision is that many studies have pointed out the influence of gender differences on the intestinal microbiome and our focus of research is not to explore the impact of gender differences on the results of the experiment. Therefore, to rule out the possible influence of sex difference, we all selected male mice as experimental subjects.

c: Response to comments : Why was the number of mice chosen used? Was a power analysis or other statistical analysis done to ensure proper numbers of mice?

Response: Thanks for your comment. For the number of mice, we did not do a power analysis to determine it before the experiment. However, we fully considered that only enough samples can reflect the overall level of change, and considering the ability of our laboratory staff, we finally chose the scheme of 10 mice in each group, a total of 30 mice.

Special thanks to you for your good comments.

Reviewer #2: 

a: Response to comments: Table 1: I think that the readability of this table could be increased as follow: mean and SD could be put together (mean�SD) and you should add the p-value. Moreover, you can add a legend for this table.

Response: Thank you for your advice on our Table, which makes it easier to read and clearer. We revised the expression of mean and SD and added P-values between the groups in the revised manuscript.

b: Response to comments: Lines 165-167: you could also provide data for differences between N and HF.

Response: Thanks for your suggestion. We added the P-value between groups at the end of the sentence.

c: Response to comments: Line 48: the term “asth ation” is unclear. Please amend it.

Many times (i.e. line 164, 181, 193, 206, etc.), in the paper there is reported the following “Error! Reference source not found”, in correspondence to references to Figures and Tables citations. Please amend it.

Response: Thanks for your comment. We apologize for the spelling and formatting errors in the original version, all of which have been corrected in the new version.

Special thanks to you for your good comments.

Reviewer #3: 

a: Response to comments: Line 48, the reviewer is not familiar with asth ation, can the authors clarify what this word is and its meaning. Perhaps it is a typo.

Response: We apologize for the spelling and formatting errors in the original version, all of which have been corrected in the new version.

b: Response to comments: The introduction provides information about some background, but it is jumbled and would benefit from a reorganization and rewrite. Further there does not seem to be an overt purpose or hypothesis established for the study.

Response: Thank you for your comments. We rearrange the paragraphs in introduction, and describe the purpose and results of our experiment.

c: Response to comments: The mice need a more complete description and location of the vendor purchased from. Microbiome/microbiota can vary greatly between vendors and thus influence results of the study.

Response: Thank you for your comments. We clarified the mice strains and the location of the vendor in the revised manuscript. The laboratory animal center of Chongqing Medical university is the experimental animal center of Chongqing National Biological Industry Base and the experimental animal service platform of National Innovative Drug Incubation (Chongqing) Base. By the Chongqing Laboratory Animal Quality Inspection Center and the Chongqing Science and Technology Commission, the center have obtained the production and animal use license of SPF-grade rats and mice, and the animal use license of rabbits, guinea pigs, pigs, dogs, sheep, monkeys, and birds, etc. The quality of experimental animals can be fully guaranteed.

d: Response to comments: Given the perceived purpose is to examine the influence of fiber and asthma, the experimental design seems incomplete, there is no asthma-fiber group. Only normal, asthma and high fiber.

Response: Thank you for your comments. We mentioned the grouping of animals in line 91-92, “the 30 mice were divided into the normal group (N1-N10), asthma group (AS1-AS10), and high-fiber diet group (HF1-HF10).”, and the establishment of models in different groups in line 99-100, “All animals except normal mice were sensitized and challenged with ovalbumin (OVA) to induce airway allergic inflammation.” The HF group was established by high fiber diet intervention in the establishment period of the asthma model.

e: Response to comments: There does not appear to the tables for dietary composition. How were diets altered to account for the high cellulose content? Protein, fat? There was a lot of time spent on the importance of fat in the introduction. This would be critical for understanding interpretation of the results.

Response: Thank you for your suggestion. For the explanation of the feed formula, we put it in lines 96-97, but it lacks the explanation of other important ingredients in feed. Therefore, we decided to upload the complete formula of feed to the journal as a supplement material in the revised manuscript.

f: Response to comments: If all animals were challenged with OVA, then should the groups be renamed?

Response: Thanks for your comment. The grouping and modeling have been explained before. In order to make it easier for readers to understand different groups, we named the normal group N group, the asthma group AS group, and the asthma group with high fiber diet intervention, and we named it HF group based on the main intervention measures.

g: Response to comments: The authors do not appear to have a scoring outline in place to understand how the IHC data was analyzed to note differences in staining.

Response: Thanks for your comment. In order to determine the infiltration of inflammatory cells in the lungs of mice, we used H&E staining instead of IHC, and then observed the morphology and structure of bronchioles and the changes of infiltration of inflammatory cells around them under light microscope.

h: Response to comments: The authors have 16S rDNA in the V4-V5 regions, this should be rRNA. While one must extract the DNA it then must be prepared for rRNA analysis.

Response: Thanks for your comment. We corrected the inaccurate expression in the method section

i: Response to comments: SCFA in animal models are analyzed using cecal contents so that fecal pellets can be used for microbiota.

Response: Thanks for your comment. We know that SCFAs are mainly absorbed in the cecum and enter the blood circulation. The content of cecum can reflect the changes of SCFAs in mice more directly. In our experiment, considering that SCFAs may be a potential biomarker in human body, and the most convenient way to detect SCFAs in human body is to detect the content of SCFAs in fresh feces. So in order to simulate this situation, we collected fresh feces of mice in the experiment, and immediately put them into the refrigerator at - 80 ℃ for storage, to minimize the possible volatilization of SCFAs.

j: Response to comments: The statistical analysis is very vague and does not employ appropriate techniques to analyze m data of this fashion.

Response: Thanks for your comment. We apologize for the vague expression in the data analysis section. In the revised manuscript, we redescribed different data analysis methods for different experimental contents.

Special thanks to you for your good comments.

---

## [Decision Letter · Decision Letter 1]

7 Jul 2021

PONE-D-21-07750R1

High Cellulose dietary intake relieves asthma inflammation through the intestinal microbiome in a mouse model

PLOS ONE

Dear Dr. Li,

Thank you for submitting your manuscript to PLOS ONE. After careful consideration, we feel that it has merit but does not fully meet PLOS ONE’s publication criteria as it currently stands. Therefore, we invite you to submit a revised version of the manuscript that addresses the points raised during the review process.

More specifically, small changes are still needed.

We look forward to receiving your revised manuscript.

Kind regards,

Catherine Mounier

Academic Editor

PLOS ONE

Journal Requirements:

Additional Editor Comments (if provided):

Reviewers' comments:

Reviewer's Responses to Questions

**Comments to the Author**

1. If the authors have adequately addressed your comments raised in a previous round of review and you feel that this manuscript is now acceptable for publication, you may indicate that here to bypass the “Comments to the Author” section, enter your conflict of interest statement in the “Confidential to Editor” section, and submit your "Accept" recommendation.

Reviewer #4: All comments have been addressed

2. Is the manuscript technically sound, and do the data support the conclusions?

Reviewer #4: Partly

3. Has the statistical analysis been performed appropriately and rigorously? 

Reviewer #4: Yes

4. Have the authors made all data underlying the findings in their manuscript fully available?

Reviewer #4: Yes

5. Is the manuscript presented in an intelligible fashion and written in standard English?

Reviewer #4: No

6. Review Comments to the Author

Reviewer #4: There are many small problems in the article which need to be carefully revised by the author.Authors should check their manuscripts carefully before submitting them

(1)The author still hasn't corrected the "See Fig Error!".

(2)The methods are not very detailed, such as how long it takes to build the mouse model and how long the mice will be intervening.

(3)The picture is very blurry.

7. PLOS authors have the option to publish the peer review history of their article (what does this mean?). If published, this will include your full peer review and any attached files.

Reviewer #4: No

---

## [Author Response · Author response to Decision Letter 1]

12 Jul 2021

Responds to the reviewer`s comments:

Reviewer #4: There are many small problems in the article which need to be carefully revised by the author. Authors should check their manuscripts carefully before submitting them

(1)The author still hasn't corrected the "See Fig Error!".

Response: Thanks for your comment. We apologize for the errors in the manuscript, and we have corrected the formatting errors that have already been mentioned.

(2)The methods are not very detailed, such as how long it takes to build the mouse model and how long the mice will be intervening.

Response: Thanks for your comment. We added a detailed description of the steps to establish the mouse model in the method section.

(3)The picture is very blurry.

Response: Thanks for your comment. We replaced the unclear figures and adjusted them by Preflight Analysis and Conversion Engine (PACE).

Special thanks to you for your good comments.

---

## [Editor Report · Decision Letter 2]

25 Nov 2021

PONE-D-21-07750R2High Cellulose dietary intake relieves asthma inflammation through the intestinal microbiome in a mouse modelPLOS ONE

Dear Dr. Li,

Thank you for submitting your manuscript to PLOS ONE. After careful consideration, we feel that it has merit but does not fully meet PLOS ONE’s publication criteria as it currently stands. Therefore, we invite you to submit a revised version of the manuscript that addresses the points raised during the review process. I still observe a major problem with the figure which are impossible to read.

Please increase the quality of the picture. This is absolutely necessary before the final acceptation

We look forward to receiving your revised manuscript.

Kind regards,

Catherine Mounier

Academic Editor

PLOS ONE
---

## [Author Response · Author response to Decision Letter 2]

15 Dec 2021

Based on the instructions provided in your letter, we replaced all the figure files of the manuscript. We sincerely hope that the images uploaded this time will meet the requirements for publication in your magazine.

---

## [Editor Report · Decision Letter 3]

27 Jan 2022

High Cellulose dietary intake relieves asthma inflammation through the intestinal microbiome in a mouse model

PONE-D-21-07750R3

Dear Dr. Li,

We’re pleased to inform you that your manuscript has been judged scientifically suitable for publication and will be formally accepted for publication once it meets all outstanding technical requirements.

Kind regards,

Catherine Mounier

Academic Editor

PLOS ONE
---

## [Editor Report · Acceptance letter]

1 Mar 2022

PONE-D-21-07750R3 

High Cellulose dietary intake relieves asthma inflammation through the intestinal microbiome in a mouse model 

Dear Dr. Li:

I'm pleased to inform you that your manuscript has been deemed suitable for publication in PLOS ONE. Congratulations! Your manuscript is now with our production department. 

Kind regards, 

on behalf of

Dr. Catherine Mounier 

Academic Editor

PLOS ONE